# Electrosynthesis of buckyballs with fused-ring systems from PCBM and its analogue

Wei-Feng Wang [1], Kai-Qing Liu [1], Chuang Niu [1], Yun-Shu Wang [2], Yang-Rong Yao [3], Zheng-Chun Yin [1], Muqing Chen [3,4], Shi-Qi Ye [1], Shangfeng Yang [3] ✉ & Guan-Wu Wang [1,5] ✉

[6,6]-Phenyl-$C_{61}$-butyric acid methyl ester (PCBM), a star molecule in the fullerene field, has found wide applications in materials science. Herein, electrosynthesis of buckyballs with fused-ring systems has been achieved through radical α-C−H functionalization of the side-chain ester for both PCBM and its analogue, [6,6]-phenyl-$C_{61}$-propionic acid methyl ester (PCPM), in the presence of a trace amount of oxygen. Two classes of buckyballs with fused bi- and tricyclic carbocycles have been electrochemically synthesized. Furthermore, an unknown type of a bisfulleroid with two tethered [6,6]-open orifices can also be efficiently generated from PCPM. All three types of products have been confirmed by single-crystal X-ray crystallography. A representative intramolecularly annulated isomer of PCBM has been applied as an additive to inverted planar perovskite solar cells and boosted a significant enhancement of power conversion efficiency from 15.83% to 17.67%.

Esters are ubiquitous and an important class of organic compounds, which have been widely applied in the chemical and pharmaceutical industries, perfumery and cosmetics, and serve as precursors for many other types of organic molecules. Esters can undergo various functional group interconversions. It is also well known that the most common α-C−H functionalizations of esters are realized through enolate formation with a base followed by alkylation, acylation, or addition[1]. In addition, the α-arylation of esters can be achieved by the reactions of enolate intermediates with aryl halides under the catalysis of transition metals[2–4]. However, the α-C−H functionalizations of esters via a radical process are rarely investigated. The in situ formed enolates transfer a single electron to oxidants such as ferrocenium and copper salts to generate persistent radical intermediates, which can undergo further reactions[5,6].

On the other hand, various functionalized derivatives of buckyballs have attracted extensive attention due to their excellent performances in materials and biological science[7–11]. [6,6]-Phenyl-$C_{61}$-butyric acid methyl ester (PCBM, **1**), a star molecule in fullerene chemistry and materials science, has been widely used in solar cell devices[12–16] due to its unique advantages in excellent solubility and electronic transmission capability. Derivatives of PCBM have also been synthesized and utilized in organic photovoltaics (OPVs)[12,17,18] and perovskite solar cells (PSCs)[13–16,18]. For example, bis-PCBM and PCBM($CH_2$) were applied in organic solar cells with P3HT as the electron donor. As desired, both P3HT:bis-PCBM and P3HT:PCBM($CH_2$) cells showed higher open circuit voltage ($V_{oc}$) and power conversion efficiency (PCE) than the P3HT:PCBM cell (0.76 V and 0.73 V vs 0.61 V; 4.4% and 4.6% vs 4.0%, respectively)[19]. The methano $CH_2$ moiety in PCBM($CH_2$) was introduced first through the reaction of [60]fullerene ($C_{60}$) with $^i$PrO-Me$_2$SiCH$_2$MgCl followed by Cu(II)-promoted cyclization to provide $C_{60}$($CH_2$). Then, the reaction of $C_{60}$($CH_2$) with the diazo compound generated from p-tosylhydrazide and NaOMe afforded PCBM($CH_2$) as a mixture of regioisomers in an overall yield of 15%[20]. Isomer effects on the device performance of OPVs and PSCs have been demonstrated[18].

[1]Hefei National Research Center for Physical Sciences at the Microscale and Department of Chemistry, University of Science and Technology of China, Hefei, Anhui 230026, P. R. China. [2]Hefei No. 1 High School, Hefei, Anhui 230601, P. R. China. [3]Hefei National Research Center for Physical Sciences at the Microscale, CAS Key Laboratory of Materials for Energy Conversion, and Department of Materials Science and Engineering, University of Science and Technology of China, Hefei, Anhui 230026, P. R. China. [4]School of Environment and Civil Engineering, Dongguan University of Technology, Dongguan, Guangdong 523808, P. R. China. [5]State Key Laboratory of Applied Organic Chemistry, Lanzhou University, Lanzhou, Gansu 730000, P. R. China. ✉e-mail: sfyang@ustc.edu.cn; gwang@ustc.edu.cn

Taking the previous complex synthetic steps for an isomeric mixture of PCBM(CH$_2$)[20] into consideration and in continuation of our interest in the electrosynthesis of fullerene derivatives[21–26], we anticipated that the straightforward cyclopropanation of the electrochemically generated dianion of PCBM with CH$_2$I$_2$ via dual nucleophilic reactions would regioselectively afford a single isomer of PCBM(CH$_2$). Surprisingly, CH$_2$I$_2$ was not involved, and two classes of buckyballs with fused bi- and tricyclic carbocycles via intramolecular radical α-C−H functionalization of the ester moiety were isolated for both PCBM and its analogue, i.e., [6,6]-phenyl-C$_{61}$-propionic acid methyl ester (PCPM, **2**)[27]. The formed buckyballs fused with a [3-6]/[3-5] bicycle or a [3-6-3]/[3-5-3] tricycle (vide infra) have different fused-ring systems compared to the previously reported buckyballs with a [5-6]/[5-5] bicycle[28,29] or a [3-7-3]/[5-5-5] tricycle[30,31]. Furthermore, an unknown type of bisfulleroid with two tethered [6,6]-open orifices could be obtained for PCPM. Herein, we describe the details of these intramolecular annulations of the electrochemically generated dianions of PCBM and PCPM in the presence of a trace amount of oxygen, and preliminarily investigate the application of a representative derivative in inverted planar PSC devices.

## Results

The reaction of **1**$^{2-}$, which was generated by controlled potential electrolysis (CPE) of **1** at −1.38 V vs. saturated calomel electrode (SCE) for ca. 1 h, with CH$_2$I$_2$ for 5 h failed to afford the desired PCBM(CH$_2$) and instead provided a buckyball with fused bicyclic carbocycle **3** as the major product and its regioisomer **4** as the minor product under an argon atmosphere. Products **3** and **4** were intramolecularly annulated isomers of PCBM. It turned out that the trace amount of O$_2$ (15.3 ppm) in Ar was crucial for the formation of **3** and **4**. Products **3** and **4** were formed in 25% and 6% yields, respectively, along with another buckyball with fused tricyclic carbocycle **5** as a minor product in 5% yield after stirring for 5 h at a flow rate of 2.6 mL s$^{-1}$ for Ar/O$_2$ gas. Notably, the electrochemically intramolecular radical α-C−H annulation of the ester moiety has been achieved. A fullerenyl proton existed in product **3** and was believed to come from protonation of an annulated anionic intermediate. Therefore, trifluoroacetic acid (TFA) was added to achieve higher protonation efficiency. Satisfactorily, the yields of **3** and **4** were improved to 39% and 12%, respectively, after the generated **1**$^{2-}$ was stirred for another 5 h under an Ar/O$_2$ atmosphere, followed by treatment with 1 equiv. of TFA for 10 min (Fig. 1a). Simultaneously, tricyclic carbocycle **5** could be obtained as a minor product in 6% yield. Further study showed that similar results were obtained without acid treatment, probably because the protonation process occurred during column chromatography on silica gel[32]. Products **3**, **4**, and **5** could be obtained in lower yields of 31%, 6%, and 4%, respectively, when 1 equiv. of TFA was immediately added after the theoretical number of coulombs required for full conversion of **1** into **1**$^{2-}$ was reached, indicating that the intermediates for **3**, **4**, and **5** were formed during electrolysis and continued to be generated upon stirring for another 5 h.

Subsequently, the corresponding electrochemical reaction of PCPM (**2**), an analogue of PCBM, was also explored. Detailed investigations demonstrated that buckyballs with fused bi- and tricycles **6** and **7** could be formed in 11% and 14% yields, respectively, after the generated **2**$^{2-}$ was stirred for another 3 h without acid treatment (Fig. 1b). Compared to those of **3**, **4**, and **5**, the fused bi- and tricycles of **6** and **7** shrank from the [3-6] and [3-6-3] patterns to the [3-5] and [3-5-3] patterns, respectively. Interestingly, product **6** could be selectively isolated in a higher yield of 32% when the generated **2**$^{2-}$ was immediately treated with 1 equiv. of TFA for 10 min (Fig. 1b). These results indicated that the electrochemical reaction of **2**$^{2-}$ under the Ar/O$_2$ atmosphere initially afforded an intermediate for **6**, which was then further transformed into a species for **7** upon prolonging the reaction time.

Remarkably, bisfulleroid **8** (vide infra) could be produced in 14% yield together with **6** in 21% yield when the generated **2**$^{2-}$ was stirred for another 3 h under an Ar/O$_2$ atmosphere and subsequently treated with 1 equiv. of TFA for 10 min (Fig. 1c). Intriguingly, bisfulleroid **8** has two tethered orifices at [6,6]-bonds with a 14-membered-ring opening instead of the previous common bisfulleroids at [5,6]-bonds[30,33–35], exhibiting a unique addition pattern for open-cage fullerene derivatives. Further study showed that bisfulleroid **8** could be more selectively and efficiently formed in 43% yield if anionic species **2**$^{2.5-}$ was generated by CPE with the acceptance of 2.5 electrons per molecule and stirred for another 5 h under an Ar/O$_2$ atmosphere, followed by treatment with 5 equiv. of TFA for 10 min (Fig. 1c). The above results hinted that prolonging the reaction time gradually led to the formation of tricycle **7** and then bisfulleroid **8**. Further experiments demonstrated that **7** could undergo highly efficient electrochemical conversion to **8** in 72% yield, suggesting that **7** was the precursor of **8** (Fig. 1d).

It was found that the amount of O$_2$ in the system, which was measured by a trace oxygen analyser, played a crucial role in the present electrochemical reactions. Taking the electrosynthesis of bisfulleroid **8** from PCPM as an example, the product yield was lowered from 43% to 39% and 30% when the O$_2$ content in the argon gas was changed from 15.3 ppm to 3.8 ppm and 27.0 ppm, respectively, while maintaining a flow rate at 2.6 mL s$^{-1}$. The product yield dropped to 32% when the flow rate was reduced from 2.6 mL s$^{-1}$ to 1.3 mL s$^{-1}$, and the O$_2$ content of 15.3 ppm in the Ar gas remained unchanged. Thus, a smaller amount of O$_2$ in Ar (3.8 ppm) and a lower gas flow rate (1.3 mL s$^{-1}$) resulted in lower product yields, while a larger amount of O$_2$ in Ar (27.0 ppm) also led to reduced product yield with more unidentified residues. These results suggested that an appropriate amount of O$_2$ was very important to control the electrochemical reactions and achieve high yields of the desired products. It should be noted that all efforts failed for the electrosynthesis of the analogous bisfulleroid from PCBM. The exact reason for this phenomenon is currently unclear.

Products **3**−**8** were unambiguously characterized by MALDI-TOF MS, $^1$H NMR, $^{13}$C NMR, FT-IR and UV/Vis spectra. The structural assignments were further confirmed by the single-crystal X-ray structures of **3**, **4**, **7**, and **8**. Single crystals of **3** were obtained through slow evaporation of a saturated solution of **3** in carbon disulfide (CS$_2$) at ambient temperature. As shown in Fig. 2a, the structure of **3** unambiguously reveals that a [3,6]-fused bicycle is joined at the C1, C2, and C3 atoms and that a hydrogen is appended to the C4 atom. The α-carbon of the side-chain ester is bonded to the C3 atom. The bond lengths of C1−C2, C2−C3, and C3−C4 are 1.590, 1.554, and 1.587 Å, respectively, and are within the range of typical functionalized C−C bond lengths of the C$_{60}$ cage. Interestingly, the two methylenes in the fused 6-membered ring are twisted. Luckily, as the stereoisomer of **3**, single crystals of **4** showing an opposite orientation of the ester moiety (Fig. 2b) were obtained through slow evaporation of its solution in a mixture of toluene and decapyrrylcorannulene (DPC)[36] at room temperature. Single crystals of **7** were obtained by slow evaporation of a mixture of a saturated solution containing product **7** in CS$_2$ and a saturated solution of DPC in toluene at about 0 °C. As seen in Fig. 2c, a [3,5,3]-fused tricycle is joined at the C1, C2, C3, and C4 atoms. In addition, the α-carbon of the ester moiety is connected to both C3 and C4 atoms. The bond lengths for C1−C2, C2−C3, and C3−C4 are 1.602, 1.502, and 1.573 Å, respectively. Black block crystals of **8** were obtained via slow diffusion of hexane into a CS$_2$ solution of **8**. Figure 2d displays the single-crystal structure of **8**. This clearly shows that the α-carbon of the ester group is anchored to the C3−C4 bond, which is broken after addition. The nonbonded C3···C4 distance is 2.452 Å. The bond length of C2−C3 is 1.355 Å, hinting that it is a C=C double bond. The C1−C2 bond is also ruptured, and the C1···C2 separation is 2.452 Å. These two [6,6]-open fulleroid motifs of **8** make its largest orifice a 14-membered ring (Supplementary Fig. 60). Both C1 and C4 bearing hydrogen are notably uplifted from the spherical surface because of their sp$^3$ characteristics.

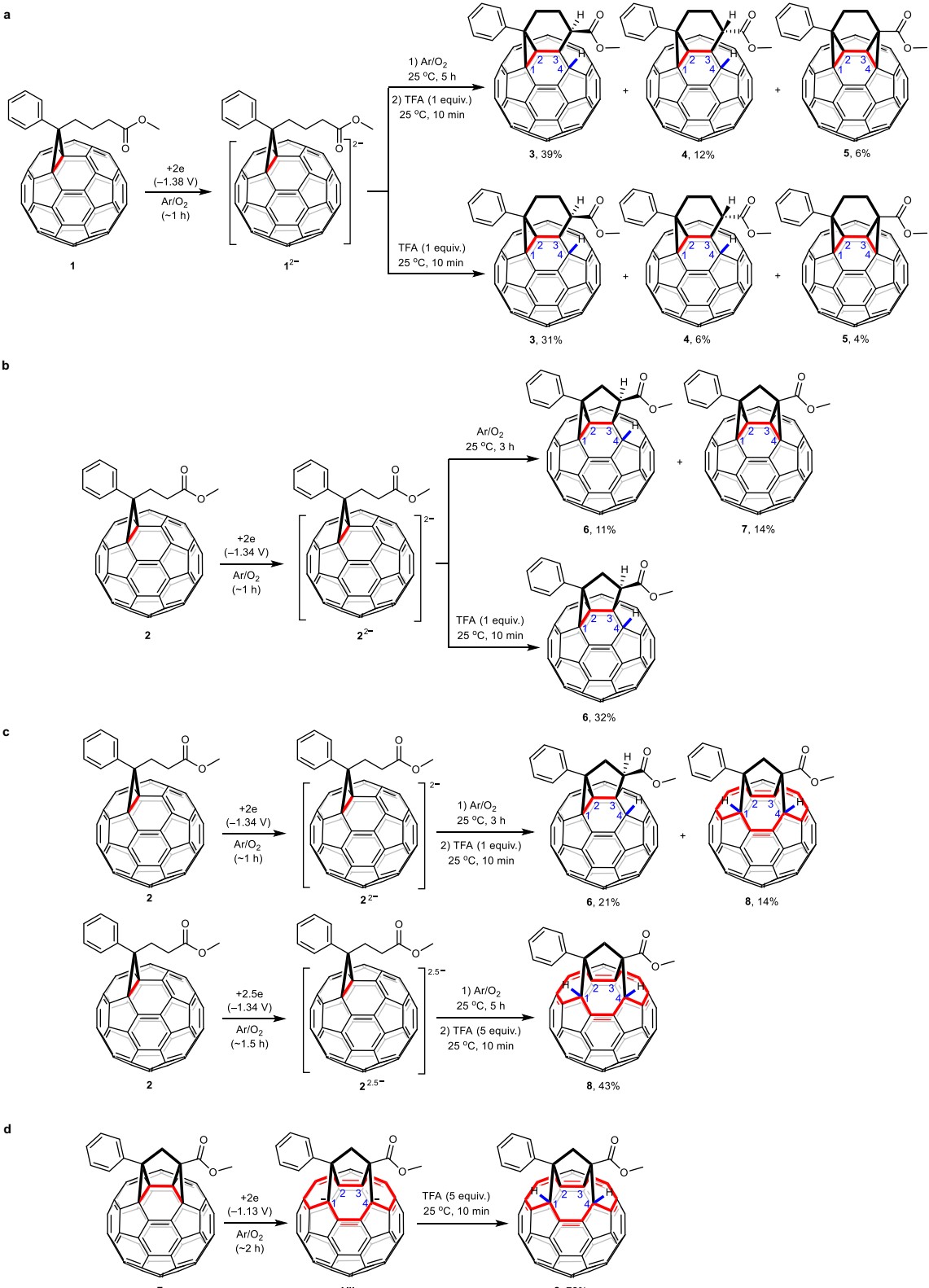

**Fig. 1 | Electrosynthesis of buckyballs. a** Electrosynthesis of buckyballs with fused bi- and tricycles **3**, **4**, and **5** from PCBM. **b** Electrosynthesis of buckyballs with fused bi- and tricycles **6** and **7** from PCPM. **c** Electrosynthesis of buckyball with fused bicycle **6** and bisfulleroid **8** from PCPM. **d** Electrochemical conversion of **7** to **8**. All

the redox potentials mentioned are relative to saturated calomel electrode. PCBM: **1**, [6,6]-phenyl-C$_{61}$-butyric acid methyl ester. PCPM: **2**, [6,6]-phenyl-C$_{61}$-propionic acid methyl ester. TFA: trifluoroacetic acid. The red color indicates the characteristic addition site for the fused ring system or the 14-membered-ring opening.

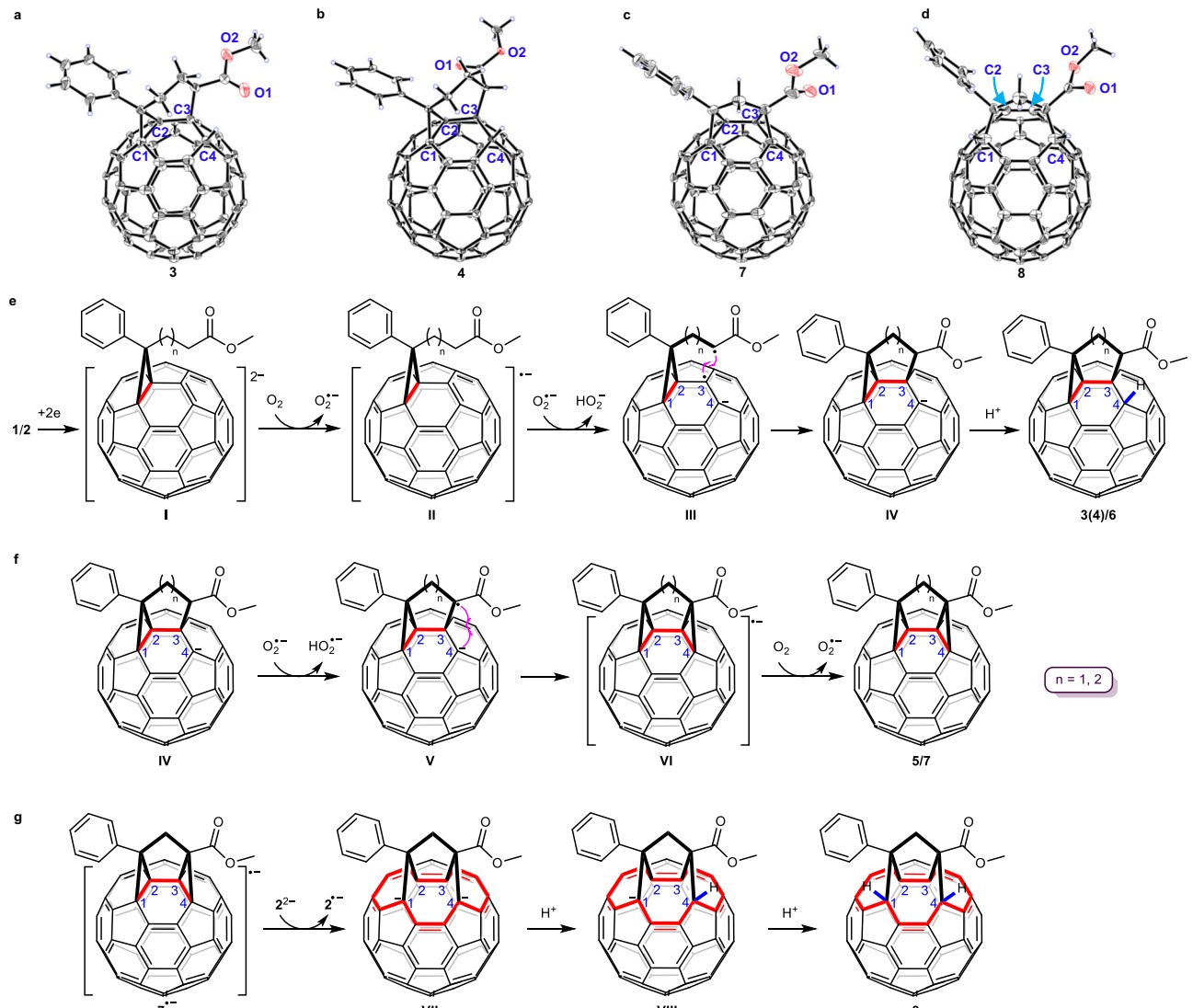

**Fig. 2 | Oak Ridge thermal ellipsoid plot (ORTEP) diagrams and proposed reaction mechanisms.** a–d ORTEP diagrams of **3**, **4**, **7**, and **8** with 20%, 20%, 10%, and 20% thermal ellipsoids, respectively. The solvent and decapyrrylcorannulene molecules are omitted for clarity. e–g Proposed reaction mechanisms for the three types of fullerene products.

On the basis of the above experimental results, plausible reaction mechanisms are proposed and shown in Fig. 2e–g. **1/2** is electro-reduced by CPE to form dianion **I** (**1**$^{2-}$/**2**$^{2-}$), which undergoes single electron transfer (SET) to O$_2$ to produce radical anion **II** and O$_2$$^{\bullet-}$. Hydrogen abstraction of the sp$^3$ α-C−H adjacent to the ester group by O$_2$$^{\bullet-}$ generates radical species **III** and HO$_2$$^-$. Subsequent intramolecular radical coupling gives fullerenyl anion **IV** with a fused bicyclic carbocycle. Final protonation affords the buckyball with fused bicycle **3**(**4**)/**6** (Fig. 2e). Upon prolonging the reaction time, another hydrogen abstraction of the sp$^3$ α-C−H adjacent to the ester group in **IV** by O$_2$$^{\bullet-}$ generates intermediate **V**, which undergoes intramolecular cyclization to result in radical anion **VI**. Finally, oxidation of **VI** by O$_2$ provides a buckyball with fused tricyclic carbocycle **5/7** (Fig. 2f). Alternatively, hydrogen abstraction of the sp$^3$ α-C−H and subsequent oxidation of the fullerenyl carboanion would generate a diradical species, which undergoes intramolecular radical coupling to give **5/7**. As shown in Fig. 1a, **3**, **4**, and **5** were obtained 31%, 6%, and 4% yields, respectively, when the reaction mixture was immediately treated with TFA after the theoretical number of coulombs required for full conversion of **1** into its dianion was reached, suggesting that intermediates **IV**, **V**, and **VI** should be produced from **1**$^{2-}$ triggered by a trace amount of oxygen

during electrolysis. This deduction can logically explain the failure for the attempted synthesis of PCBM(CH$_2$) from **1**$^{2-}$ due to its preferential conversion into the abovementioned intermediates during electro-lysis. For the formation of **8**, intermediate **7**$^{\bullet-}$ (**VI** when $n=1$) can be reduced to dianion **VII**, most likely by **2**$^{2-}$. The fact that the second redox potential of **2** is much more negative than the first redox potential of **7** (Supplementary Figs. 2 and 11) supports the proposed SET process. Dual protonations of **VII** via monoanion **VIII** provide bisfulleroid **8** (Fig. 2g).

To better understand the observed stereochemistries and regioselectivities, computational studies with PCPM and PCBM as the starting materials by density functional theory (DFT) calculations at the B3LYP/6-31 G(d) level were performed. The partial natural bond orbital (NBO) charge distributions of fullerenyl anion **IV** showed that C4 (−0.128 for $n=1$; −0.127 for $n=2$) was the most negatively charged carbon atom among the nonfunctionalized C$_{60}$ carbon atoms and was thus more prone to protonation, followed by C16 (−0.086 for $n=1$; −0.087 for $n=2$) as the second most negative site (Supplementary Fig. 61). Product **3** was slightly more stable than **4** by 1.03 kcal mol$^{-1}$. As a result, both **3** and **4** were formed, and **3** was preferentially generated, consistent with the experimentally observed **3**:**4** ratio of

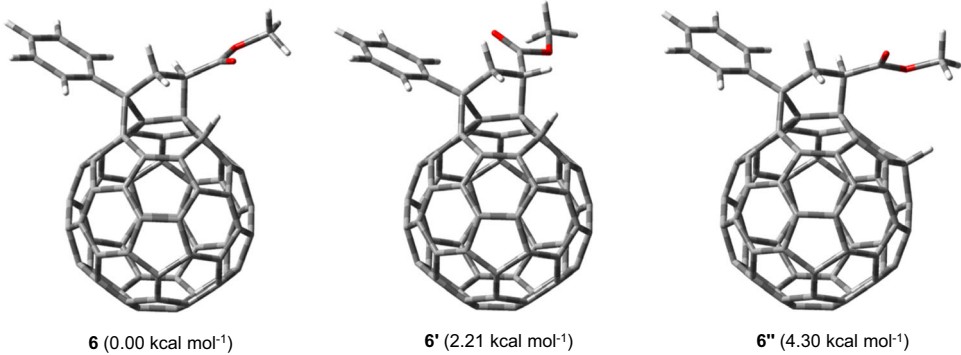

**6** (0.00 kcal mol⁻¹)          **6'** (2.21 kcal mol⁻¹)          **6"** (4.30 kcal mol⁻¹)

**Fig. 3 | Relative energies for optimized 6, 6' and 6".** Computations were performed at the B3LYP/6-31 G(d) level.

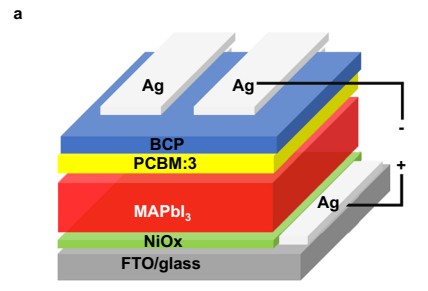

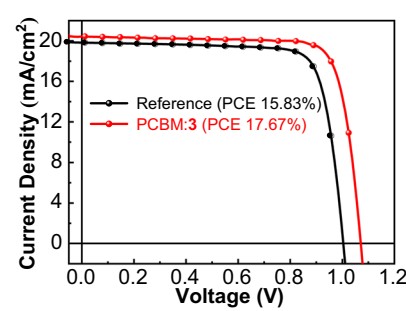

**Fig. 4 | Application to perovskite solar cells. a** Illustration of the inverted planar perovskite solar cell structure used in this work. **b** $J$–$V$ curves (reverse scan) of the device with PCBM:**3** (red line) and the reference device with PCBM (black line). BCP:

bathocuproine, 2,9-dimethyl-4,7-diphenyl-1,10-phenanthroline. PCBM: [6,6]-phenyl-$C_{61}$-butyric acid methyl ester. MA: methylammonium. FTO: fluorine doped tin oxide.

3.3–5.2. Nevertheless, the calculated energy of **6** was lower than that of **6'** by 2.21 kcal mol⁻¹ (Fig. 3), agreeing with the isolation of only **6** (see Supplementary Section 11). In addition, **6** was calculated to be more stable than isomeric **6"** by 4.30 kcal mol⁻¹. Notably, the structural optimization of dianion **7²⁻** automatically resulted in cleavage of both the C1–C2 bond and C3–C4 bond and generated open-cage intermediate **VII**. The NBO charge distributions of dianion **VII** exhibited the four most negative sites at C1 (−0.082), C4 (−0.075), C10 (−0.076), and C16 (−0.073). The corresponding protonated monoanion **VIII** was more stable than other isomeric monoanions by at least 3.46 kcal mol⁻¹. As a result, **VIII** was favorably formed. Similarly, the NBO charge distributions of **VIII** showed the three most negative sites at C1 (−0.073), C10 (−0.076), and C12 (−0.049). Theoretical calculations indicated that the protonated product **8** was more stable than isomers **8'** and **8"** by at least 7.26 kcal mol⁻¹. These computational results provide rationales for the experimentally obtained stereochemistries and regioselectivities.

The redoxes and energy levels of products **1**–**8** are summarized in Supplementary Table 1. In particular, the calculated lowest unoccupied molecular orbital (LUMO) and the highest occupied molecular orbital (HOMO) energy levels of **3** were −3.70 and −5.40 eV, which were comparable to those (−3.72 and −5.43 eV) of PCBM. Compared to PCBM, adducts **3**–**8** have limited solubility and are not suitable to serve as electron acceptors in organic solar cells or as electron transport layer (ETL) materials in PSCs. Given the wide application of fullerene derivatives in PSCs[13–18,24,25,37], the presentative product **3**, an intramolecularly annulated isomer of PCBM, was successfully applied as an additive to inverted planar PSCs with the configuration of FTO/NiOₓ/ MAPbI₃/PCBM:**3**/BCP/Ag (Fig. 4a). A reference device without **3** was also fabricated for comparison. After incorporation of **3** into

the PCBM layer, the PCE was enhanced from 15.83% to 17.67% (increased by 11.6%, Fig. 4b) owing to a dramatically increased $V_{oc}$ from 1.00 V to 1.07 V, an improved short-circuit current ($J_{sc}$) from 19.85 mA/cm² to 20.42 mA/cm² and a higher fill factor (FF) from 79.46% to 80.80%. Electron transport materials play an important role in the performance of PSCs. We, therefore, investigated the effect of the ETL film without and with incorporation of **3** on the electron mobility ($\mu_e$) for the electron-only device structures of ITO/MAPbI₃/PCBM/Ag and ITO/MAPbI₃/PCBM:**3**/Ag by using the space charge limited current (SCLC) method[38]. The electron mobility for the device with incorporation of **3** was $1.40 \times 10^{-3}$ cm² V⁻¹ S⁻¹. Noticeably, this value was nearly twice as high as that ($7.16 \times 10^{-4}$ cm² V⁻¹ S⁻¹) of the reference device, resulting in an increase in $V_{oc}$, $J_{sc}$, and FF and thus PCE[39,40]. These results demonstrated that isomer **3** had enormous potential in solar cell devices.

## Discussion
In summary, the electrosynthesis of buckyballs with fused-ring systems has been achieved via intramolecular radical α-C–H functionalization of the side-chain ester for both PCBM and its analogue PCPM in the presence of a trace amount of oxygen. Two classes of buckyballs with fused bicycles or tricycles can be electrochemically synthesized. Furthermore, an unknown type of bisfulleroid with two tethered [6,6]-open orifices can be efficiently obtained from PCPM. The chemical structures for all three types of products have been established by single-crystal X-ray crystallography. An intramolecularly annulated isomer of PCBM has been applied as an additive to inverted planar perovskite solar cells, and the power conversion efficiency increases significantly from 15.83% to 17.67%, demonstrating its enormous potential in solar cell devices.

## Methods

### Generation of $1^{2-}$ and subsequent treatment with $CH_2I_2$

Dianionic $1^{2-}$ was electrochemically obtained from **1** (9.2 mg, 0.010 mmol) by CPE at −1.38 V vs. SCE for ca. 1 h, followed by treatment with $CH_2I_2$ (0.80 µL, 0.010 mmol) and stirring at 25 °C for 5 h. All operations were performed under an argon atmosphere (15.3 ppm $O_2$) and the gas flow rate was 2.6 mL s⁻¹. The resulting mixture was purified on a silica gel column (200–300 mesh) with $CS_2$/$CH_2Cl_2$ (1:1 v/v) to afford a mixture (3.3 mg) containing products **3** (2.27 mg, 25%), **4** (0.55 mg, 6%), and **5** (0.48 mg, 5%) based on the integrals of the methoxy group in the ¹H NMR spectrum (Supplementary Fig. 52). The mixture was further separated by high-performance liquid chromatography (HPLC) on a Cosmosil Buckyprep column (10 × 250 mm) using toluene/isopropanol (7:3 v/v) as the eluent with a flow rate of 3 mL min⁻¹ to obtain pure products **3**, **4**, and **5**.

### Generation of $1^{2-}$ and subsequent treatment with TFA

Dianionic $1^{2-}$ was electrochemically obtained from **1** (9.2 mg, 0.010 mmol) by CPE at −1.38 V vs. SCE for ca. 1 h, and then stirred at 25 °C for 5 h, followed by treatment with trifluoroacetic acid (TFA, 0.76 µL, 0.010 mmol) for 10 min. All operations were performed under an argon atmosphere, in which the $O_2$ content was determined to be 15.3 ppm by a trace oxygen analyser, and the gas flow rate was 2.6 mL s⁻¹. The resulting mixture was purified on a silica gel column (200–300 mesh) with $CS_2$/$CH_2Cl_2$ (1:1 v/v) to afford a mixture (5.3 mg) containing products **3** (3.63 mg, 39%), **4** (1.11 mg, 12%), and **5** (0.56 mg, 6%) based on the integrals of the methoxy group in the ¹H NMR spectrum (Supplementary Fig. 53). The mixture was further separated by HPLC on a Cosmosil Buckyprep column (10 × 250 mm) using toluene/isopropanol (7:3 v/v) as the eluent at a flow rate of 3 mL min⁻¹ to afford pure products **3**, **4**, and **5**.

Dianionic $1^{2-}$ was obtained by electroreduction from **1** (8.9 mg, 0.010 mmol) at −1.38 V vs. SCE by CPE, and the electrolysis time lasted for about 1 h. The reaction mixture was immediately treated with TFA (0.76 µL, 0.010 mmol) for 10 min. All operations were performed under an argon atmosphere (15.3 ppm $O_2$), and the gas flow rate was 2.6 mL s⁻¹. The resulting mixture was purified on a silica gel column (200–300 mesh) with $CS_2$/$CH_2Cl_2$ (1:1 v/v) to afford a mixture (7.0 mg) containing products **3** (2.79 mg, 31%), **4** (0.55 mg, 6%), and **5** (0.32 mg, 4%) along with recovered **1** (3.34 mg, 38%) based on the integrals in the ¹H NMR spectrum (Supplementary Fig. 54). The mixture was further separated by HPLC on a Cosmosil Buckyprep column (10 mm × 250 mm) using toluene/isopropanol (7:3 v/v) as the eluent with a flow rate of 3 mL min⁻¹ to obtain pure products **3**, **4**, and **5**.

### Generation of $2^{2-}$ and subsequent treatment without or with TFA

Dianionic $2^{2-}$ was electrochemically obtained from **2** (9.2 mg, 0.010 mmol) by CPE at −1.34 V vs. SCE for ca. 1 h and then stirred at 25 °C for 3 h. All operations were performed under an argon atmosphere (15.3 ppm $O_2$), and the gas flow rate was 2.6 mL s⁻¹. The resulting mixture was filtered and then purified on a silica gel column (300–400 mesh) with $CS_2$ as the eluent to give products **6** (1.0 mg, 11%) and **7** (1.3 mg, 14%).

Dianionic $2^{2-}$ was electrochemically obtained from **2** (8.8 mg, 0.010 mmol) by CPE at −1.34 V vs. SCE for ca. 1 h, followed by immediate treatment with TFA (0.76 µL, 0.010 mmol) for 10 min. All operations were performed under an argon atmosphere (15.3 ppm $O_2$), and the gas flow rate was 2.6 mL s⁻¹. The resulting mixture was filtered and then purified on a silica gel column (300–400 mesh) with $CS_2$ as the eluent to give product **6** (2.8 mg, 32%).

Dianionic $2^{2-}$ was electrochemically obtained from **2** (8.9 mg, 0.010 mmol) by CPE at −1.34 V vs. SCE for ca. 1 h and then stirred at 25 °C for 3 h, followed by treatment with TFA (0.76 µL, 0.010 mmol) for 10 min. All operations were performed under an argon atmosphere

(15.3 ppm $O_2$), and the gas flow rate was 2.6 mL s⁻¹. The resulting mixture was purified on a silica gel column (200–300 mesh) with $CS_2$/$CH_2Cl_2$ (1:1 v/v) to afford a mixture containing products **6** and **8**, which was further separated by HPLC on a Cosmosil Buckyprep column (10 × 250 mm) using toluene as the eluent with a flow rate of 3 mL min⁻¹. From the ¹H NMR spectrum of the mixture (3.1 mg), **6** (1.87 mg, 21%) and **8** (1.23 mg, 14%) were afforded as amorphous brown solids (Supplementary Fig. 55).

### Synthesis of product 8

Dianionic $2^{2.5-}$ was electrochemically obtained from **2** (8.8 mg, 0.010 mmol) by CPE at −1.34 V vs. SCE for ca. 1.5 h and then stirred at 25 °C for 5 h, followed by treatment with TFA (3.8 µL, 0.050 mmol) for 10 min. All operations were performed under an argon atmosphere (15.3 ppm $O_2$), and the gas flow rate was 2.6 mL s⁻¹. The resulting mixture was filtered and then purified on a silica gel column (300–400 mesh) with $CS_2$ as the eluent to give product **8** (3.8 mg, 43%).

## Data availability

All the data supporting the findings of this study are provided in the Supplementary Information and Source Data files. Crystallographic data for the structures reported in this article have been deposited at the Cambridge Crystallographic Data Center under deposition numbers CCDC 2235544 (**3**), 2263924 (**4**), 2235545 (**7**) and 2235547 (**8**). Copies of the data can be obtained free of charge via https://www.ccdc.cam.ac.uk/structures/. Cartesian coordinates and energies of the calculated structures are available in the provided source data file. All other data are available from the corresponding author upon request. Source data are provided with this paper.

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

## Acknowledgements

We are grateful for financial support from the National Natural Science Foundation of China (No. 22071231 to G.-W.W. and No. 22201278 to C.N.) and the Guangdong Basic and Applied Basic Research Foundation (No. 2022A1515140089 to M.C.). The theoretical calculations reported in this paper were performed at the Supercomputing Center of the University of Science and Technology of China. We appreciate Prof. Tong-Xin Liu of Henan Normal University for providing DPC and Prof. Xin Hong of Zhejiang University for helpful discussion.

## Author contributions

G.-W.W. and S.Y. supervised the project. G.-W.W. and K.-Q.L. conceived the study. W.-F.W., Y.-S.W., and S.-Q.Y. performed the experiments, fabricated the PSC devices and measured their PCEs and analysed all the data. C.N., Y.-R.Y., Z.-C.Y., and M.C. characterized the X-ray structures of four compounds. Z.-C.Y. helped the theoretical calculations. G.-W.W. and W.-F.W. wrote the paper. All authors discussed the results and commented on the paper.

## Competing interests

The authors declare no competing interests.
