## [Peer Review File · Nature Communications]

Electrosynthesis of buckyballs with fused-ring systems from PCBM and its analogueReviewers' Comments:

Reviewer #1:

Remarks to the Author:

The authors of this manuscript achieved the electrosynthesis of a series of C60 derivatives with fused-ring systems through radical α -C-H functionalization of the side-chain ester for PCBM and PCPM, in the presence of a trace amount of oxygen. The reaction mechanism of the new C60 derivatives was analyzed, and their molecular structures were carefully characterized and confirmed by single-crystal X-ray crystallography. An isomer 3 of PCBM was used as an additive in the PCBM electron-transport layer (ETL) to the n-i-p type planar perovskite solar cells (pero-SCs), and they found that the device with 3 as additive of the ETL demonstrated a higher power conversion efficiency (PCE) of 17.67%, while the PCE of the control device is only 15.83%. The results are interesting for the related researchers. Therefore I think this manuscript can be accepted for publication after some minor revisions as indicated in the following:

- (1) When the authors introduce the C60 derivative PCPM (Compound 2), the following related literature reporting its optoelectronic properties should be cited for the easy reference of readers: G. Zhao, et al., "Effect of carbon chain length in the substituent of PCBM-like molecules on their photovoltaic properties", *Adv. Funct. Mater.* 2010, 20, 1480-1487.
- (2) When the authors mention the electrosynthesis condition (controlled potential), for example in the first sentence in "Results and discussion": "The reaction of 12-, which was generated by controlled potential electrolysis (CPE) of 1 at -1.38 V for ca. 1 h", the reference electrode should be indicated, "at -1.38 V" should be revised into "at -1.38 V vs. SCE".
- (3) For the application of Compound 3 as additive in the PCBM ETL, the electron mobility, the lowest unoccupied molecular orbital (LUMO) and the highest occupied molecular orbital (HOMO) energy levels of Compound 3 should be reported, since these properties are very important for the application as ETL and for explaining the origin of the higher Voc of the device based on the ETL with the additive.

Reviewer #2:

Remarks to the Author:

This paper presents the preparation, characterization, and photovoltaic properties of new fullerene derivatives prepared electrochemically. First of all, fullerene derivatives including PCBM are no longer star molecules. Non-fullerene acceptors have emerged very rapidly and the power conversion efficiency (PCE) exceeds 19%, which is much higher than the PCE of 10-11% with PCBM. Second, the enhancement of PCE by 1-2% in perovskite solar cells have been frequently seen by the use of various additives. Thus, such marginal improvement is one of many analogous examples. Finally, the authors claimed novelty and importance in the electrochemical synthesis of C60 with fused-ring systems. Although they are unprecedented in a narrow range, similar cis-1 adducts are known and this work is the simple, routine extension of addressing fullerene reactivity by authors and others. Therefore, I regret to say that the paper does not reach the high-level of standard warranting the publication in *Nature Communications*.

Reviewer #3:

Remarks to the Author:

This paper reported "new" but "undesired" products of buckyballs (C60) with fused-ring system in well-known [6,6]-Phenyl-C61-butyric acid methyl ester (PCBM) and [6,6]-phenyl-C61-propionic acid methyl ester (PCPM) using electrochemical approach as intermediate process. The yield of the products was depended on the presence of trace amount oxygen and mixing time during protonation reaction with trifluoroacetic acid (TFA) to form fused bicyclic and tricyclic carbocycles. Formation of bisfulleroid derivatives was also achieved utilizing fused tricyclic product from PCPM. All products are well characterized supporting the proposed reaction mechanism. Product with fused bicyclic system

derived from PCBM was utilized as additive in inverted planar perovskite solar cells and photo-energy conversion efficiency (PCE) enhancement was reported.

Even though, there are many reported fullerenes derivatives fused to organic systems. It is worth to note that this is the first reported PCBM / PCPM derivatives with fused-ring system that synthesized via electrochemical approach as intermediate reaction process. However, there are some critical points that could be fixed to further improve the paper as follows;

1. In the Introduction part it was mentioned that this research was originally attempted to electro synthesize PCBM(CH₂) as author claims due to its "simplicity" compared with already reported works [1]. It turned out that such reaction produces "undesired" buckyballs (C₆₀) with fused-ring system product. The author should explain more in scientific point of view, why electrochemical approach is chosen. Also, it should be explained why single isomer of PCBM(CH₂) is failed to be synthesized via electrochemical approach. This paper lacks of explanation about this point.

2. In regards to point 1, it seems to the reader that this paper only report and discuss mainly about the "undesired product" which is irrelevant with the main purpose mentioned above. Author should add more relevant background and reference about fullerene derivatives with fused ring systems and redefine/rewrite the background including the importance of this research.

3. The author should give a reason why product 3 is decided as representative material for additive in inverted planar perovskite solar cell. Why not as an acceptor in organic solar cell? etc. Also, it is worth to explain the mechanism behind the PCE enhancement when product 3 is applied as additive, addressing the presence of fused bicyclic system in PCBM. Please write the experimental condition and setup to obtain the resulting JV curve

In conclusion, this paper needs major correction before publication. Defining clear scientific background, importance, and motivation of research is vital to elaborate with relevant results and discussion.

Reference

[1] Li, C.-Z., Chien, S.-C., Yip, H.-L., Chueh, C.-C., Chen, F.-C., Matsuo, Y., Nakamura, E. & Jen, A. K.-Y. Facile synthesis of a 56π-electron 1,2-dihydromethano-[60]PCBM and its application for thermally stable polymer solar cells. *Chem. Commun.* 47, 10082–10084 (2011).

REVIEWER COMMENTS

Reviewer #1 (Remarks to the Author):

The authors of this manuscript achieved the electrosynthesis of a series of C60 derivatives with fused-ring systems through radical α -C-H functionalization of the side-chain ester for PCBM and PCPM, in the presence of a trace amount of oxygen. The reaction mechanism of the new C60 derivatives was analyzed, and their molecular structures were carefully characterized and confirmed by single-crystal X-ray crystallography. An isomer 3 of PCBM was used as an additive in the PCBM electron-transport layer (ETL) to the n-i-p type planar perovskite solar cells (pero-SCs), and they found that the device with 3 as additive of the ETL demonstrated a higher power conversion efficiency (PCE) of 17.67%, while the PCE of the control device is only 15.83%. The results are interesting for the related researchers. Therefore I think this manuscript can be accepted for publication after some minor revisions as indicated in the following:

(1) When the authors introduce the C60 derivative PCPM (Compound 2), the following related literature reporting its optoelectronic properties should be cited for the easy reference of readers:

G. Zhao, et al., "Effect of carbon chain length in the substituent of PCBM-like molecules on their photovoltaic properties", *Adv. Funct. Mater.* 2010, 20, 1480-1487.

(2) When the authors mention the electrosynthesis condition (controlled potential), for example in the first sentence in "Results and discussion": "The reaction of 12-, which was generated by controlled potential electrolysis (CPE) of 1 at -1.38 V for ca. 1 h", the reference electrode should be indicated, "at -1.38 V" should be revised into "at -1.38 V vs. SCE".

(3) For the application of Compound 3 as additive in the PCBM ETL, the electron mobility, the lowest unoccupied molecular orbital (LUMO) and the highest occupied molecular orbital (HOMO) energy levels of Compound 3 should be reported, since these properties are very important for the application as ETL and for explaining the origin of the higher V_{oc} of the device based on the ETL with the additive.

Reviewer #2 (Remarks to the Author):

This paper presents the preparation, characterization, and photovoltaic properties of new fullerene derivatives prepared electrochemically. First of all, fullerene derivatives including PCBM are no longer star molecules. Non-fullerene acceptors have emerged very rapidly and the power conversion efficiency (PCE) exceeds 19%, which is much higher than the PCE of 10-11% with PCBM. Second, the enhancement of PCE by 1-2% in perovskite solar cells have been frequently seen by the use of various additives. Thus, such marginal improvement is one of many analogous examples. Finally, the authors claimed novelty and importance in the electrochemical synthesis of C60 with fused-ring systems. Although they are unprecedented in a narrow range, similar cis-1 adducts are known and this work is the simple, routine extension of addressing fullerene reactivity by authors and others. Therefore, I regret to say that the paper does not reach the high-level of standard warranting the publication in *Nature Communications*.

Reviewer #3 (Remarks to the Author):

This paper reported “new” but “undesired” products of buckyballs (C₆₀) with fused-ring system in well-known [6,6]-Phenyl-C₆₁-butyric acid methyl ester (PCBM) and [6,6]-phenyl-C₆₁-propionic acid methyl ester (PCPM) using electrochemical approach as intermediate process. The yield of the products was depended on the presence of trace amount oxygen and mixing time during protonation reaction with trifluoroacetic acid (TFA) to form fused bicyclic and tricyclic carbocycles. Formation of bisfulleroid derivatives was also achieved utilizing fused tricyclic product from PCPM. All products are well characterized supporting the proposed reaction mechanism. Product with fused bicyclic system derived from PCBM was utilized as additive in inverted planar perovskite solar cells and photo-energy conversion efficiency (PCE) enhancement was reported.

Even though, there are many reported fullerenes derivatives fused to organic systems. It is worth to note that this is the first reported PCBM / PCPM derivatives with fused-ring system that synthesized via electrochemical approach as intermediate reaction process. However, there are some critical points that could be fixed to further improve the paper as follows;

1. In the Introduction part it was mentioned that this research was originally attempted to electro synthesize PCBM(CH₂) as author claims due to its “simplicity” compared with already reported works [1]. It turned out that such reaction produces “undesired” buckyballs (C₆₀) with fused-ring system product. The author should explain more in scientific point of view, why electrochemical approach is chosen. Also, it should be explained why single isomer of PCBM(CH₂) is failed to be synthesized via electrochemical approach. This paper lacks of explanation about this point.
2. In regards to point 1, it seems to the reader that this paper only report and discuss mainly about the “undesired product” which is irrelevant with the main purpose mentioned above. Author should add more relevant background and reference about fullerene derivatives with fused ring systems and redefine/rewrite the background including the importance of this research.
3. The author should give a reason why product 3 is decided as representative material for additive in inverted planar perovskite solar cell. Why not as an acceptor in organic solar cell? etc. Also, it is worth to explain the mechanism behind the PCE enhancement when product 3 is applied as additive, addressing the presence of fused bicyclic system in PCBM. Please write the experimental condition and setup to obtain the resulting JV curve

In conclusion, this paper needs major correction before publication. Defining clear scientific background, importance, and motivation of research is vital to elaborate with relevant results and discussion.

Reference

- [1] Li, C.-Z., Chien, S.-C., Yip, H.-L., Chueh, C.-C., Chen, F.-C., Matsuo, Y., Nakamura, E. & Jen, A. K.-Y. Facile synthesis of a 56 π -electron 1,2-dihydromethano-[60]PCBM and its application for thermally stable polymer solar cells. *Chem. Commun.* 47, 10082–10084 (2011).

Point-by-point responses to reviewers' comments

Reviewer #1 (Remarks to the Author):

The authors of this manuscript achieved the electrosynthesis of a series of C60 derivatives with fused-ring systems through radical α -C-H functionalization of the side-chain ester for PCBM and PCPM, in the presence of a trace amount of oxygen. The reaction mechanism of the new C60 derivatives was analyzed, and their molecular structures were carefully characterized and confirmed by single-crystal X-ray crystallography. An isomer **3** of PCBM was used as an additive in the PCBM electron-transport layer (ETL) to the n-i-p type planar perovskite solar cells (pero-SCs), and they found that the device with **3** as additive of the ETL demonstrated a higher power conversion efficiency (PCE) of 17.67%, while the PCE of the control device is only 15.83%. The results are interesting for the related researchers. Therefore I think this manuscript can be accepted for publication after some minor revisions as indicated in the following:

(1) When the authors introduce the C60 derivative PCPM (Compound **2**), the following related literature reporting its optoelectronic properties should be cited for the easy reference of readers:

G. Zhao, et al., "Effect of carbon chain length in the substituent of PCBM-like molecules on their photovoltaic properties", *Adv. Funct. Mater.* 2010, 20, 1480-1487.

Response: We thank the reviewer for bringing up this point. We have now cited this literature as Ref. 27.

(2) When the authors mention the electrosynthesis condition (controlled potential), for example in the first sentence in "Results and discussion": "The reaction of **12-**, which was generated by controlled potential electrolysis (CPE) of **1** at -1.38 V for ca. 1 h", the reference electrode should be indicated, "at -1.38 V" should be revised into "at -1.38 V vs. SCE".

Response: We have adopted the reviewer's suggestion. We have added "vs. saturated calomel electrode (SCE)" after "at -1.38 V" and added "vs. SCE" after other potentials throughout the whole manuscript.

(3) For the application of Compound **3** as additive in the PCBM ETL, the electron mobility, the lowest unoccupied molecular orbital (LUMO) and the highest occupied molecular orbital (HOMO) energy levels of Compound **3** should be reported, since these properties are very important for the application as ETL and for explaining the origin of the higher V_{oc} of the device based on the ETL with the additive.

Response: We appreciate the reviewer's insightful comment. To address the reviewer's concern, we have added "In particular, the calculated lowest unoccupied molecular orbital (LUMO) and the highest occupied molecular orbital (HOMO) energy levels of **3** were -3.70 and -5.40 eV, which were comparable to those (-3.72 and -5.43 eV) of PCBM." and "Electron transport materials play an important role in the performance of PSCs. We therefore investigated the effect of the ETL film without and with incorporation of **3** on the electron mobility (μ_e) for the electron-only device structures of ITO/CH₃NH₃PbI₃/PCBM/Ag and ITO/CH₃NH₃PbI₃/PCBM:**3**/Ag by using the space

charge limited current (SCLC) method.³⁸ The electron mobility for the device with incorporation of **3** was $1.40 \times 10^{-3} \text{ cm}^2 \text{ V}^{-1} \text{ S}^{-1}$. Noticeably, this value was nearly twice as high as that ($7.16 \times 10^{-4} \text{ cm}^2 \text{ V}^{-1} \text{ S}^{-1}$) of the reference device, resulting in an increase in J_{sc} , V_{oc} and FF and thus PCE.^{39,40}. Refs. 39 and 40 have been cited to support the statement.

Reviewer #2 (Remarks to the Author):

This paper presents the preparation, characterization, and photovoltaic properties of new fullerene derivatives prepared electrochemically. First of all, fullerene derivatives including PCBM are no longer star molecules. Non-fullerene acceptors have emerged very rapidly and the power conversion efficiency (PCE) exceeds 19%, which is much higher than the PCE of 10-11% with PCBM. Second, the enhancement of PCE by 1-2% in perovskite solar cells have been frequently seen by the use of various additives. Thus, such marginal improvement is one of many analogous examples. Finally, the authors claimed novelty and importance in the electrochemical synthesis of C60 with fused-ring systems. Although they are unprecedented in a narrow range, similar *cis*-1 adducts are known and this work is the simple, routine extension of addressing fullerene reactivity by authors and others. Therefore, I regret to say that the paper does not reach the high-level of standard warranting the publication in Nature Communications.

Response: We thank the reviewer for the comments regarding our work. The essential discovery of our work was the formation of buckyballs with fused-ring systems from the electrochemical reaction of PCMB and its analogue PCPM through the unprecedented radical α -C-H functionalization of their linear side chain. The application of a representative product **3** (most abundant among the derived products) as an additive to the PCBM electron transport layer (ETL) was just to demonstrate its utility. PCBM has been widely used in organic photovoltaics (OPVs) and perovskite solar cells (PSCs) and is still a star molecule in the fullerene field. We specified PCMB as a star molecule to emphasize its importance in the fullerene field, rather than the whole science/material field. If the reviewer is still strongly opposed to this statement, we can delete it. The use of product **3** as an additive to the PCBM ETL could improve the PCE from 15.83% to 17.67%. We have added “increased by 11.6%” in the parenthesis immediately after “from 15.83% to 17.67%” to show the increased percentage. Indeed, there are similar *cis*-1 adducts with fused ring systems in the literature. The fused-ring systems that are mostly related to our work are buckyballs fused with a bicycle or a tricycle, yet with different ring sizes. To address the reviewer’s raised issue, we have added “The formed buckyballs fused with a [3-6]/[3-5] bicycle or a [3-6-3]/[3-5-3] tricycle (vide infra) have different fused-ring systems compared to the previously reported buckyballs with a [5-6]/[5-5] bicycle^{28,29} or a [3-7-3]/[5-5-5] tricycle.^{30,31}” and cited the four most closely related references.

Reviewer #3 (Remarks to the Author):

This paper reported “new” but “undesired” products of buckyballs (C60) with fused-ring system in well-known [6,6]-Phenyl-C61-butyric acid methyl ester (PCBM) and [6,6]-phenyl-C61-propionic acid methyl ester (PCPM) using electrochemical approach as intermediate process. The yield of the products was depended on the presence of trace

amount oxygen and mixing time during protonation reaction with trifluoroacetic acid (TFA) to form fused bicyclic and tricyclic carbocycles. Formation of bisfulleroid derivatives was also achieved utilizing fused tricyclic product from PCPM. All products are well characterized supporting the proposed reaction mechanism. Product with fused bicyclic system derived from PCBM was utilized as additive in inverted planar perovskite solar cells and photo-energy conversion efficiency (PCE) enhancement was reported.

Even though, there are many reported fullerenes derivatives fused to organic systems. It is worth to note that this is the first reported PCBM / PCPM derivatives with fused-ring system that synthesized via electrochemical approach as intermediate reaction process. However, there are some critical points that could be fixed to further improve the paper as follows;

1. In the Introduction part it was mentioned that this research was originally attempted to electro synthesize PCBM(CH₂) as author claims due to its “simplicity” compared with already reported works [1]. It turned out that such reaction produces “undesired” buckyballs (C₆₀) with fused-ring system product. The author should explain more in scientific point of view, why electrochemical approach is chosen. Also, it should be explained why single isomer of PCBM(CH₂) is failed to be synthesized via electrochemical approach. This paper lacks of explanation about this point.

Response: We appreciate the reviewer’s thoughtful advice. To make the proposed idea clear, we have changed “we attempted the electrochemical reaction of commercially available PCBM with CH₂I₂ in an effort to regioselectively obtain a single isomer of PCBM(CH₂).” to “we anticipated that the straightforward cyclopropanation of the electrochemically generated dianion of PCBM with CH₂I₂ via dual nucleophilic reactions would regioselectively afford a single isomer of PCBM(CH₂).”. To explain why the synthesis of a single isomer of PCBM(CH₂) failed via an electrochemical approach, we have added “As shown in Fig. 1a, **3**, **4** and **5** were obtained in 31%, 6% and 4% yields, respectively, when the reaction mixture was immediately treated with TFA after the theoretical number of coulombs required for full conversion of **1** into its dianion was reached, suggesting that intermediates **IV**, **V** and **VI** should be produced from **1**²⁻ triggered by a trace amount of oxygen during electrolysis. This deduction can logically explain the failure for the attempted synthesis of PCBM(CH₂) from **1**²⁻ due to its preferential conversion into the abovementioned intermediates during electrolysis.” during the discussion of the reaction mechanism.

2. In regards to point 1, it seems to the reader that this paper only report and discuss mainly about the “undesired product” which is irrelevant with the main purpose mentioned above. Author should add more relevant background and reference about fullerene derivatives with fused ring systems and redefine/rewrite the background including the importance of this research.

Response: We thank the reviewer for the constructive suggestion. Indeed, there are some fullerene derivatives with fused ring systems. The fused-ring systems that are mostly related to our work are buckyballs fused with a bicycle or a tricycle, yet with different ring sizes. To address the reviewer’s raised issue, we have added “The formed buckyballs fused with a [3-6]/[3-5] bicycle or a [3-6-3]/[3-5-3] tricycle (vide infra) have different fused-ring systems compared to the previously reported buckyballs with a [5-6]/[5-5] bicycle^{28,29} or a [3-7-3]/[5-5-5] tricycle.^{30,31}” and cited the four most closely related references.

3. The author should give a reason why product **3** is decided as representative material for additive in inverted planar perovskite solar cell. Why not as an acceptor in organic solar cell? etc. Also, it is worth to explain the mechanism behind the PCE enhancement when product **3** is applied as additive, addressing the presence of fused bicyclic system in PCBM. Please write the experimental condition and setup to obtain the resulting JV curve.

Response: We appreciate the reviewer's constructive comment. Product **3** has similar HOMO and LUMO energy levels as PCBM (see Supplementary Table 5). Product **3** has a structure with a fused bicyclic system and is the most abundant product derived from PCBM but has limited solubility. Therefore, we decided to employ **3** as an additive in the PCBM ETL layer. To address the reviewer's concerns, we have added "In particular, the calculated lowest unoccupied molecular orbital (LUMO) and the highest occupied molecular orbital (HOMO) energy levels of **3** were -3.70 and -5.40 eV, which were comparable to those (-3.72 and -5.43 eV) of PCBM. Compared to PCBM, adducts **3-8** have limited solubility and are not suitable to serve as electron acceptors in organic solar cells or as electron transport layer (ETL) materials in PSCs." We have also added "Electron transport materials play an important role in the performance of PSCs. We therefore investigated the effect of the ETL film without and with incorporation of **3** on the electron mobility (μ_e) for the electron-only device structures of ITO/CH₃NH₃PbI₃/PCBM/Ag and ITO/CH₃NH₃PbI₃/PCBM:**3**/Ag by using the space charge limited current (SCLC) method.³⁸ The electron mobility for the device with incorporation of **3** was $1.40 \times 10^{-3} \text{ cm}^2 \text{ V}^{-1} \text{ S}^{-1}$. Noticeably, this value was nearly twice as high as that ($7.16 \times 10^{-4} \text{ cm}^2 \text{ V}^{-1} \text{ S}^{-1}$) of the reference device, resulting in an increase in J_{sc} , V_{oc} and FF and thus PCE.^{39,40}" (Refs. 39 and 40 were cited to support the statement) to explain the mechanism behind the PCE enhancement. In addition, the experimental conditions and setup to obtain the resulting $J-V$ curve have been added to the Supplementary Information with the paragraph "The current density-voltage ($J-V$) characterizations were performed with a Keithley 2400 source meter under simulated AM 1.5 irradiation (100 mW cm^{-2}) with a standard xenon-lamp-based solar simulator (SAN-EI-XES-50S2). The simulator illumination intensity was calibrated with a monocrystalline silicon reference cell (Newport P/N 91150 V, with KG-5 visible color filter) calibrated by the National Renewable Energy Laboratory (NREL).".

In conclusion, this paper needs major correction before publication. Defining clear scientific background, importance, and motivation of research is vital to elaborate with relevant results and discussion.

Reference

[1] Li, C.-Z., Chien, S.-C., Yip, H.-L., Chueh, C.-C., Chen, F.-C., Matsuo, Y., Nakamura, E. & Jen, A. K.-Y. Facile synthesis of a 56π -electron 1,2-dihydromethano-[60]PCBM and its application for thermally stable polymer solar cells. Chem. Commun. 47, 10082–10084 (2011).

Response: We are deeply thankful to the reviewer's insightful comments. We have revised our manuscript accordingly to improve its quality. We have cited the mentioned reference.

Reviewers' Comments:

Reviewer #1:

Remarks to the Author:

The authors of the manuscript have revised their manuscript according to the reviewers' comments and revision opinions. Now the revised manuscript can be accepted for publication at its present form.

Reviewer #3:

Remarks to the Author:

The authors have adequately addressed my comments and I support publication.

REVIEWER COMMENTS

Reviewer #1 (Remarks to the Author):

The authors of the manuscript have revised their manuscript according to the reviewers' comments and revision opinions. Now the revised manuscript can be accepted for publication at its present form.

Reviewer #3 (Remarks to the Author):

The authors have adequately addressed my comments and I support publication.

Point-by-point responses to reviewers' comments

Reviewer #1 (Remarks to the Author):

The authors of the manuscript have revised their manuscript according to the reviewers' comments and revision opinions. Now the revised manuscript can be accepted for publication at its present form.

Response: We appreciate the reviewer's effort in improving the quality of our manuscript.

Reviewer #3 (Remarks to the Author):

The authors have adequately addressed my comments and I support publication.

Response: We appreciate the reviewer's effort in improving the quality of our manuscript.